# Intergenerational Perioperative Neurocognitive Disorder

**DOI:** 10.3390/biology12040567

**Published:** 2023-04-07

**Authors:** Ling-Sha Ju, Timothy E. Morey, Christoph N. Seubert, Anatoly E. Martynyuk

**Affiliations:** 1Department of Anesthesiology, College of Medicine, University of Florida, P.O. Box 100254, JHMHC, 1600 SW Archer Road, Gainesville, FL 32610, USA; 2Brain Institute, College of Medicine, University of Florida, Gainesville, FL 32610, USA

**Keywords:** perioperative neurocognitive disorder, traumatic brain injury, sevoflurane, heritable perioperative neurocognitive disorder

## Abstract

**Simple Summary:**

Hundreds of millions of patients of all ages undergo general anesthesia for surgery or periods of sedation each year, making the adverse effects of general anesthesia (GA)/surgery a public health concern of paramount importance. The accelerated neurocognitive decline after GA/surgery, also known as perioperative neurocognitive disorder (PND), is a widely recognized public health problem. Advanced age, which tracks with an increasing prevalence of elevated stress, inflammation, and neurodegenerative diseases, is the most consistent contributing factor in the development of PND. Although young adults are more resilient to PND, such resilience may be weakened in specific groups of young adults with pathophysiological conditions characterized by excessive or chronic stress and inflammation. In this narrative review, we discuss the roles of stress, inflammation, and epigenetic changes in the development of PND and experimental findings demonstrating that the effects of surgery, traumatic brain injury, and the general anesthetic sevoflurane interact to induce persistent dysregulation of the stress response system, inflammation markers, and behavior in young adult male rats and in their future offspring (intergenerational PND).

**Abstract:**

Accelerated neurocognitive decline after general anesthesia/surgery, also known as perioperative neurocognitive disorder (PND), is a widely recognized public health problem that may affect millions of patients each year. Advanced age, with its increasing prevalence of heightened stress, inflammation, and neurodegenerative alterations, is a consistent contributing factor to the development of PND. Although a strong homeostatic reserve in young adults makes them more resilient to PND, animal data suggest that young adults with pathophysiological conditions characterized by excessive stress and inflammation may be vulnerable to PND, and this altered phenotype may be passed to future offspring (intergenerational PND). The purpose of this narrative review of data in the literature and the authors’ own experimental findings in rodents is to draw attention to the possibility of intergenerational PND, a new phenomenon which, if confirmed in humans, may unravel a big new population that may be affected by parental PND. In particular, we discuss the roles of stress, inflammation, and epigenetic alterations in the development of PND. We also discuss experimental findings that demonstrate the effects of surgery, traumatic brain injury, and the general anesthetic sevoflurane that interact to induce persistent dysregulation of the stress response system, inflammation markers, and behavior in young adult male rats and in their future offspring who have neither trauma nor anesthetic exposure (i.e., an animal model of intergenerational PND).

## 1. Introduction

Perioperative neurocognitive disorder (PND) is an overarching term describing cognitive impairments in patients who underwent surgery under general anesthesia (GA) [1,2,3,4,5,6,7,8,9,10]. The PND term, introduced by the International Nomenclature Consensus Working Group in 2018, covers cognitive declines diagnosed not only after GA/surgery but also cognitive deficiencies identified prior to GA/surgery [11,12]. Postoperative cognitive impairments are further segregated into postoperative delirium, delayed neurocognitive recovery, and postoperative neurocognitive disorder [11,12]. Postoperative delirium and delayed neurocognitive recovery are acute neurocognitive abnormalities diagnosed up to a week and a month post-GA/surgery, respectively, and manifest as confusion, agitation, delusions, and deficiencies in orientation and attention [11,12]. The signs of the decline of cognitive functions (i.e., memory, awareness, reasoning, judgment, language), identified by neuropsychological tests between 1 and 12 months after GA/surgery, are diagnostic criteria for a postoperative neurocognitive disorder component of PND [11,12]. PND is associated with poorer recovery, increased dependence on social assistance, and higher mortality [2,5,6,10,13,14].

The Initial mechanisms mediating PND and the relative roles of GA and surgery in inducing PND are poorly understood. Clinical and laboratory evidence suggest that advanced age, preexisting neurodegenerative diseases, surgery-induced stress, and inflammation are essential factors contributing to the development of PND [1,2,3,4,5,6,7,8,9,10,14,15,16,17,18,19,20,21,22,23,24,25,26,27]. Notably, studies in animal models demonstrate that not just surgery under GA but GAs alone may induce neurobehavioral abnormalities, lasting dysregulations of stress response systems, and upregulation of the inflammation markers (an animal model of PND) [10,18,19,20,21,22,25,28].

Because neurodegenerative diseases become more prevalent and worsen with age, it is not surprising that the aging population is more vulnerable to PND, as it is to most other diseases [14,17,29,30,31]. An important question is whether various preexisting pathophysiological conditions that involve persistently dysregulated stress response systems, inflammation, and neurological/neurocognitive deficiencies move the vulnerability to PND towards a younger age. Perhaps even more important, at least ethically, is the question of whether conditions that may lead to parental PND increase the likelihood that their unexposed offspring conceived after the parental PND-inducing events will develop similar abnormalities (intergenerational PND). 

The notion of heritable pathophysiological effects of parental exposure to persistent stress has evolved from being denied to being supported by numerous studies in animal models and humans [32,33,34,35,36,37,38,39,40,41,42,43,44,45,46,47,48,49,50,51,52,53,54,55]. Emerging laboratory studies support the possibility that by acting via mechanisms involving aberrations in stress signaling and epigenomic regulations, GAs may induce neurobehavioral abnormalities not only in the exposed animals but also in their future unexposed offspring [56,57,58,59,60]. Remarkably, experimental findings show that GABAergic GAs may induce heritable effects when administered from the early postnatal period to at least young adulthood, thus covering nearly all age groups that typically procreate [57,58,59]. The large number of patients who require GA/surgery before or during their reproductive age, the even larger number of their future unexposed offspring whose health may be affected by parental PND, and a growing number of brain-related disorders of unknown etiology in unexposed individuals [37,38,39,40,41,42,43,44,45,46,47,48,49,50,51,52,53,54,55,61,62], in which parental experiences may be a predisposing factor, underscore the importance of investigating the heritability of PND. In humans, determining intergenerational effects of parental experiences in general, and anesthesia/surgery in particular, is complicated by confounding effects in the form of ethical, social, and environmental factors. Our findings in rats add to this complexity by demonstrating that offspring could be affected even if their parents are asymptomatic [57]. Nevertheless, the persistent and heritable adverse effects of stress in specific circumstances, for instance, large groups of people affected by war or famine in relatively compact living areas within a defined period of time, have been extensively studied [36,63,64,65,66,67,68,69,70]. The findings of such human studies—along with the findings of stress-like intergenerational effects of GAs in animal studies [57,58,59], which had the advantage of strictly controlled experiments—support the urgent need to elucidate the causes, mediating mechanisms, and biomarkers of PND in millions of patients and their potentially affected children.

The purpose of this narrative review is to draw attention to the possibility of intergenerational PND, a new phenomenon which, if confirmed in humans, may be important for public health considering the great number of people who might be affected. Here, we discuss data in the literature and the authors’ own experimental findings on stress, inflammation, and epigenetic aspects of PND, in general, and the potential role of the interaction of stress and inflammation caused by pathophysiological conditions and GA/surgery in PND development in young adults and their future offspring. In support of this concept, we put special focus on the discussion of our experimental findings demonstrating that the effects of surgery, traumatic brain injury (TBI), and the GA-sevoflurane (SEVO) interact to induce persistent dysregulation of the stress response system, increased inflammation, and behavioral deficits in young adult male rats (an animal model of PND) and in their offspring who have neither trauma nor anesthetic exposure (intergenerational PND). We further highlight that TBI/GA/surgery-induced intergenerational PND extends from neurobehavioral abnormalities to abnormal functioning of the HPA axis. This allows easily measurable, objective biomarkers of intergenerational PND, such as prepulse inhibition (PPI) of the acoustic startle response, as well as levels of corticosterone and inflammation markers in the blood that may readily be translated to human subjects. Because the PND term was introduced relatively recently [11,12], postoperative cognitive dysfunction (POCD) and PND terms will be used interchangeably throughout this review to discuss findings from earlier and more recent studies.

## 2. Epidemiology

The reported incidences of PND vary substantially, depending on the definition, patient population, surgery types, time of assessment, neuropsychological test batteries, etc. [10,13,14,17,71]. In 1955, Bedford [72] first undertook a retrospective observational study of dementia after GA. He reported that 7% of elderly patients met the criteria for extreme dementia after GA. In contrast, Simpson et al. [73] reported in 1961 that there was no significant association between anesthesia and cognitive deterioration in patients (≥65 years of age). In the following decades, many studies were devoted to proving the existence of this phenomenon. There was limited progress in studying PND, however, partially because of a lack of an accurate definition of PND and standardized neuropsychological tests [31,74]. In 1995 POCD, based on a consensus panel definition, was reported in 5% of 231 patients (≥65 years of age) at 6 months after knee replacement [75]. In 1998, a multicenter trial was undertaken by the International Study of Post-Operative Cognitive Dysfunction (ISPOCD1) to investigate the occurrence of cognitive decline in 1218 patients (≥65 years of age) after noncardiac surgery. Delayed neurocognitive recovery was found in 25.8% of patients 1 week after surgery, and POCD was found in 9.9% of patients 3 months after surgery, compared with 3.4% and 2.8% of UK controls. Increasing age and duration of anesthesia, less education, repeated surgery, postoperative infections, and respiratory complications were risk factors for cognitive dysfunction at the 1-week timepoint, while only age remained a risk factor for cognitive impairment 3 months later [30]. Using the same neuropsychological methodology as the ISPOCD1 study, Monk et al. [27] found that the incidence of delayed neurocognitive recovery in older patients (≥60 years of age) undergoing noncardiac surgery was 41.4% at hospital discharge and that of POCD was 12.7% at 3 months follow-up. According to another study, POCD persisted for up to 3 months in 56% of patients (≥64 years of age) after orthopedic surgery [76]. Newman et al. undertook a systematic review of POCD after noncardiac surgery, which included 46 articles. The review concluded that POCD was present in a significant proportion of patients in the early weeks after major noncardiac surgery, with the elderly being more at risk [31]. In another systematic review involving 24 studies, the pooled incidence of POCD at 3 months following noncardiac surgeries was 11.7% [77]. Although there is strong agreement that advancing age is a risk factor for PND development, younger patients may not be immune to the adverse neurocognitive effects of GA/surgery [1,3,4,5,6,7,18,19,27]. Thus, Monk et al. assessed neurocognitive function after noncardiac surgery in 331 young adult patients (18–39 years of age) and 378 middle-aged patients (40–59 years of age) [27]. Interestingly, at hospital discharge, young adult patients tended to exhibit greater neurocognitive deficiencies (36.6% affected) than their middle-aged counterparts (30.4% affected). This difference disappeared, however, with 5.7% and 5.6% affected young adult and middle-aged patients 3 months after surgery, respectively [27]. Specifically designed studies on how different preexisting pathophysiological conditions in younger adults affect their vulnerability to subsequent surgery/GA-induced PND would not only be beneficial for developing therapeutic approaches but will also provide new insights into mechanisms mediating PND.

## 3. Pathogenesis of PND

### 3.1. Hypothalamic-Pituitary-Adrenal Axis and PND

Stress may be defined as an organism’s response aimed at restoring homeostasis perturbed by external or internal factors, described as stressors [78,79,80]. The stress response involves complex physiological processes in the brain and the body [78,79,80,81,82,83,84]. The hypothalamic-pituitary-adrenal (HPA) axis is a major neuroendocrine system involved in the regulation of stress responsivity [79,81,82,84]. Activation of the HPA axis starts with a stressor-initiated release of the corticotropin-releasing hormone (CRH) by neurons in the medial parvocellular subdivision of the paraventricular nucleus (PVN) of the hypothalamus. CRH, delivered to the anterior pituitary gland through hypophyseal portal vessels, activates its receptors to initiate the release of adrenocorticotropic hormone (ACTH). In turn, ACTH, delivered via the systemic circulation to the adrenal cortex, induces the synthesis and release of glucocorticoid steroid hormones (GCs), i.e., cortisol in humans and corticosterone in rodents [78,79,82]. GCs insert their slow gene transcription and rapid nongenomic effects through activation of the high-affinity mineralocorticoid receptors (MRs) and low-affinity glucocorticoid receptors (GRs) distributed throughout the body, including the brain [80,81,82,85,86,87,88,89]. In comparison with GRs, MRs have about a 10 times higher affinity for GCs [80,81,82,85,86,87,88,89]. According to current understanding, the majority of MRs are occupied under basal conditions and are involved in the mediation of the initial response to stress, while GRs primarily mediate the effects of elevated levels of GCs caused by stressors [78,82,84,87,90]. GCs play important physiological roles in daily activities and sleep. They are involved in the regulation of metabolic, cardiovascular, immune, and cognitive processes, among others [91]. Different aspects of cognitive activities are modulated by the activation of MRs or GRs [92,93,94,95]. Transient GC responses to environmental perturbations are often beneficial and may lead to memory consolidation and enhancement of mental/cognitive performance [80,96]. On the other hand, excessive or persistent hyperactivity of the HPA axis increases the risk of the development of immune, metabolic, cardiovascular, neurodegenerative, and psychiatric disorders [80,84,89,97,98,99,100,101,102]. It does so, at least in part, through dysregulation of HPA axis responsiveness to subsequent stressful experiences [80,103]. Therefore, a crucial aspect of normal HPA axis functioning is a timely termination of the stress response. Such a termination, or an adaptation to the effect of a stressor, is primarily achieved via GC-induced corticoid receptor-mediated inhibition of the HPA axis activity (i.e., via negative feedback) [78,79,82]. Both MRs and GRs can be involved in the mediation of the negative feedback effect of GCs on the HPA axis [79,82,90]. GABA type A receptor (GABA_A_R)-mediated control of the CRH-releasing PVN neurons is another important mechanism regulating the HPA axis [83,104,105].

A surgical insult can act as a stressor whose potency increases with the increasing invasiveness of a surgical procedure. Thus, a systematic review and meta-analysis of 71 studies involving 2953 patients of the surgery-induced cortisol response revealed that minimally invasive surgeries were not associated with increases in cortisol [106]. On the other hand, moderately and highly invasive surgical operations were accompanied by an increase in serum cortisol levels [106]. Notably, such surgery-increased secretion of cortisol was transient. The cortisol levels peaked during the 18-h period after surgery, although increased cortisol levels were still detected up to a week after surgery [106]. In addition to surgical complexity, older age, gender (women), open surgery, and general anesthesia were associated with higher levels of cortisol after surgery [106]. Higher serum and cerebrospinal fluid cortisol levels are associated with postoperative delirium [107,108,109,110,111,112,113,114], suggesting that hyperactivity of the HPA axis could be a contributing factor to such mental disturbances. This possibility is supported by the observation that patients who had elevated levels of cortisol prior to surgery were at increased risk of developing PND [115,116,117]. For example, Kazmierski et al. [115] found an increased likelihood of postoperative delirium in patients with major depressive disorder (MDD), a disease that involves greater reactivity of the HPA axis and elevated levels of cortisol. In further support of the role of the HPA axis in PND development are the findings of Manenschijn and colleagues [117]. They reported that in patients ≥ 65 years of age, older age was associated with an increased risk of delirium. However, patients with the GR polymorphisms, specifically homozygous carriers of haplotype 4, characterized by higher GR sensitivity to GCs and, therefore, a more potent GR-mediated negative feedback effect, had a 92% decreased risk of developing delirium, regardless of age, cognition, and functional state [117]. Fang et al. [116] found that intravenous administration of a high dose of dexamethasone (2 mg/kg), a potent, long-acting synthetic GC, before induction of GA increased the incidence of POCD during the early postoperative period after microvascular decompression in patients of 40 to 60 years of age. On the other hand, Sauer et al. reported that administration of a lower dose of dexamethasone (1 mg/kg) at the induction of anesthesia did not change the incidence of delirium (14.2%) in comparison with the placebo group (14.9%). Furthermore, some studies found that preventative administration of a low dose of dexamethasone (8 mg) or the GR agonist methylprednisolone could reduce the incidence of POCD [118,119]. Altogether, these findings agree with the current view of opposing cognitive effects of low and high levels of GCs [89,100]. It is important to consider that the effects of excessive activation of the HPA axis or administration of exogenous GCs, in particular synthetic GCs, are not equivalent [85,90,120]. For example, the PVN and the anterior pituitary gland may be the primary sites of action for corticosterone and dexamethasone, respectively, to induce negative feedback on HPA axis activity [106,121].

In addition to clinical studies’ findings that surgery under general anesthesia may act as a stressor [107,108,109,110,111,112,113,114], human and animal studies provide further evidence that GAs themselves can cause a glucocorticoid response. A study measuring the cortisol response to general anesthesia in 34 children undergoing medical imaging showed a significant cortisol response during recovery from anesthesia [122]. Another study in infants reported that plasma cortisol levels rose significantly after anesthesia induction but before surgery [123]. To investigate the stress response to sedation and anesthesia in children, Hsu et al. [124] measured salivary cortisol levels in largely healthy children and found that sedation and anesthesia increased cortisol levels more than threefold for all patients, regardless of the type of procedure performed (imaging, endoscopy, and minor surgical procedures). Furthermore, patients who underwent regional (spinal or epidural) anesthesia had a 17% lower area under the curve (AUC) for serum cortisol than patients who received general anesthesia [106].

#### Stress-like Effects of GABAergic Anesthetics

Studies in rodents not only support the ability of GAs, in particular those whose action involves the enhancement of GABA_A_R signaling, to induce stress-like responses but may also suggest mediating mechanisms of such effects. For example, sevoflurane, a volatile GA whose polyvalent action involves the enhancement of GABA_A_R activity, administered to neonatal or young adult rats caused an acute increase in corticosterone secretion immediately after anesthesia, as well as long-term neurobehavioral abnormalities and dysregulation of HPA axis functioning in the form of an exacerbated corticosterone responses to stress [57,58,59,125,126,127,128]. The findings in rats support the notion that GABAergic anesthetics act via specific molecular mechanisms to induce stress-like responses rather than the notion that GA-caused increases in GC levels are the result of a systemic stress response due to inadequately controlled physiological parameters during anesthesia. Specific mechanisms may include SEVO-induced reduction in the expression of MRs and GRs and subsequent impairment of corticoid receptor-mediated negative feedback modulation of the HPA axis activity by GCs [57]. Another plausible mechanism of stress-like effects of Gas may involve GA-caused impairment of GABA_A_R-mediated inhibitory control of the HPA axis [126,127,129,130,131,132].

GABA, a major inhibitory neurotransmitter in the brain, causes inhibition of neuronal activity primarily by activating the Cl^−^ current through GABA_A_R channels [133]. In most mature neurons, the transmembrane gradient of Cl^−^, which is primarily maintained by activities of the Na^+^−K^+^−2Cl^−^ (NKCC1) Cl^−^ importer and K^+^−2Cl^−^ (KCC2) Cl^−^ exporter, is such that the equilibrium potential of Cl^−^ is near to the resting membrane potential [134,135,136,137,138]. Therefore, GABA-activated Cl^−^ current, by reducing the membrane resistance, reduces neuronal excitability by weakening the excitatory potential of the excitatory currents, resulting in suppression of neuronal activity [134,135,137]. GABA_A_R signaling imposes inhibitory control of the HPA axis by suppressing the activity of the CRH-secreting hypothalamic PVN neurons [83,104,105]. This GABA_A_R-mediated inhibitory control of the HPA axis is further strengthened by GABA_A_R enhancers, the neuroactive steroids, which are products of HPA axis activation [83,104,105]. SEVO may cause sustained stress-like responses, at least in part, by modifying the GABA_A_R-mediated inhibitory control of the HPA axis. It may do so by shifting GABA_A_R signaling toward excitatory via an increase in the *Nkcc1*/*Kcc2* ratio and by enhancing “shifted towards excitatory” GABA_A_R signaling in both neonatal [127,139,140] and young adult rats (Figure 1, authors’ unpublished observations). In support of this pathway, the NKCC1 inhibitor bumetanide or the GABA_A_R antagonist bicuculline prevented both an increase in *Nkcc1/Kcc2* ratio and corticosterone release in neonatal rats exposed to SEVO [127,139,140]. Pretreatment of young adult rats with either bumetanide or the KCC2 enhancer CLP290 prior to exposure to SEVO also prevented an increase in hippocampal *Nkcc1/Kcc2* mRNA ratio and hypothalamic *Crh* mRNA levels, reduced hippocampal *Gr* and *Mr* mRNA levels, and normalized serum corticosterone levels (Figure 1, authors’ unpublished observations). Bumetanide is an FDA-approved diuretic that is currently indicated for the treatment of edema associated with heart failure and hepatic/renal disease in adults [141]. Bumetanide was shown to have promising therapeutic effects in animal models of neurological and neurocognitive disorders associated with brain trauma, chronic pain, certain types of epilepsies, autism spectrum disorders (ASDs), and Parkinson’s disease [135,142,143,144]. Of more direct relevance to the accelerated cognitive decline during aging, which is a hallmark of PND, retrospective analyses found that patients over 65 years of age who were treated with bumetanide were less likely to be affected by Alzheimer’s disease [145,146].

However, under sustained or severe stress conditions, even in mature animals, GABA_A_R signaling may upregulate rather than downregulate PVN activity because of a shift in GABA_A_R signaling towards excitation [147,148]. Such a shift in GABA_A_R signaling may occur because of stress-induced reduction in cell surface KCC2 expression in the PVN neurons [101,147,148,149,150,151,152,153]. Stress, alcohol, nicotine, TBI, and other environmental stressors may induce such a decrease in KCC2 cell surface expression [101,147,148,149,150,151,152,153]. Such changes in adult KCC2 expression were linked to a number of neuropathophysiological conditions, including depressive-like behavior, increased alcohol dependency, and epileptic seizures [152,153,154]. SEVO, when administered to young adult rats, may induce similar changes in the cell surface KCC2 expression [57]. Thus, acutely, SEVO caused a reduction in cell surface KCC2 expression in PVN neurons of young adult male rats. Long-term, these rats exhibited exacerbated HPA axis responses to stress and neurobehavioral deficiencies [57]. Notably, less prominent changes in cell surface KCC2 expression induced by SEVO in young adult female rats were accompanied by long-term neuroendocrine and neurobehavioral abnormalities that showed the same trend but were not statistically significant [57]. These findings suggest that the threshold for similar acute KCC2 and long-term neuroendocrine/neurobehavioral effects of SEVO in young adult females is higher. Interestingly, only male rodents were tested in previous studies reporting acute stress-induced downregulation of KCC2 expression [147,155,156]. However, daily exposure to stress for 25 days was reported to reduce cell-surface KCC2 expression in the female mouse hippocampus as well [151,157]. It will be important to investigate whether young adult male and female humans differ in their susceptibility to PND.

The findings in animal models suggest that corticoid receptor antagonists may alleviate acute electroencephalographic and persistent neurobehavioral effects of GAs, providing an additional line of evidence that stress-like effects of GAs may contribute to PND-like abnormalities [126,127,129,130,131,132,140,158,159]. Thus, the prototypical non-selective GR antagonist mifepristone (RU486) and the MR antagonist RU28318 depressed electroencephalography (EEG)-detectable epileptic seizure-like activity caused by the GABAergic GA propofol in neonatal rats [131], while exogenous corticosterone administered to naive pups induced long-term synaptic abnormalities and exacerbated corticosterone responses to stress similarly to those induced by neonatal administration of propofol [130]. RU28318 also reduced the ability of SEVO to induce EEG-detectable seizure-like activity in rat pups and decreased neonatal SEVO-induced exacerbated corticosterone responses to stress and anxiety-like behavior in adulthood [126,160,161,162]. Etomidate, an intravenous GA that shares a GABA_A_R-mediated mechanism of action with propofol, but in contrast to propofol, inhibits the synthesis of corticosterone [163], only weakly increased serum corticosteroid levels and elicited far less EEG-detectable seizure-like activity in neonatal rat pups [129]. However, etomidate administered to corticosterone-pretreated rat pups further increased the total duration of seizure-like activity beyond that caused by the administration of exogenous corticosterone, suggesting that both anesthetic-altered GABA_A_R signaling and corticosterone secretion are required for these GA-induced abnormalities [129]. These findings also suggest that the effects of anesthesia and environmental stressors may be additive or synergistic to induce PND. This concept is further supported by experimental findings that adult rats exposed neonatally to SEVO or etomidate followed by a subsequent single episode of maternal separation several days later exhibited neuroendocrine/neurobehavioral abnormalities significantly greater than those exposed to only one of these two stressors [140,158,159]. The findings that the interaction of adverse effects of GAs and environmental stressors, even at a very young age, may be sufficient for triggering long-term neuroendocrine and neurobehavioral abnormalities in adulthood support a possibility that young patients with preexisting conditions may be at a higher risk of surgery/anesthesia-induced PND. Surgery and anesthesia may also increase proinflammatory cytokines, which are associated with an increased risk of developing PND, as discussed below [164,165,166,167].

### 3.2. Inflammation

The inflammatory response is a defense mechanism against pathogens and insults. However, excessive inflammation, and especially a lack of its proper resolution, can become harmful and trigger a wide range of pathophysiological conditions [168]. Growing evidence suggests that inflammation, in general, and neuroinflammation, in particular, play an essential role in the pathogenesis of PND [164,165,166,167]. It is believed that the peripheral proinflammatory cytokine cascades triggered by local surgical injury increase the blood-brain barrier (BBB) permeability through disruption of the tight junctions between endothelial cells. Consequently, peripheral inflammatory cytokines penetrate the BBB and trigger neuroinflammation, which ultimately contributes to the initiation of structural and functional alterations in the brain (i.e., PND) [164,165,166,167]. 

#### 3.2.1. Peripheral Inflammatory Response

Millions of patients worldwide undergo invasive procedures that inevitably involve tissue damage, hemorrhage, and ischemic cell death. Damaged or dying cells may initiate inflammation, for example, by passively releasing damage-associated molecular patterns (DAMPs), e.g., high-mobility group box 1 (HMGB1), S100 proteins, and heat shock proteins (HSPs) [169,170,171]. The released DAMPs activate the innate immune system by binding to pattern recognition receptors (PRRs), such as the toll-like receptor (TLR) and the receptor for advanced glycation end products (RAGE) [171,172]. One of the mechanisms mediating the immune effects of DAMPs is their binding to RAGE in bone marrow–derived monocytes (BMDMs) and initiation of second messenger cascades that activate the intracellular nuclear factor kappa B (NF-κB) signaling pathway [171,172]. Activation of the NF-κB pathway ultimately leads to the synthesis and release of various proinflammatory cytokines such as interleukin-1 beta (IL-1β), interleukin-6 (IL-6), and tumor necrosis factor-alpha (TNF-α) [173]. The proinflammatory cytokines, in turn, can further promote the secretion of DAMPs by acting via a positive feedback loop, further escalating the systemic inflammatory response [167]. Additionally, IL-1 and TNF-α can directly activate NF-κB [173].

As a prototypical DAMP, HMGB1 has particularly strong associations with cognitive decline after surgery. Clinical studies have demonstrated that patients with POCD had higher serum HMGB1 levels than patients without POCD [174,175,176]. Similarly, animal studies have shown that HMGB1 is mechanistically linked with cognitive deficits after surgery [177,178,179,180]. In a landmark experiment, a single intraperitoneal injection of recombinant HMGB1 caused memory decline in mice, as evidenced by decreased freezing time in fear conditioning tests, while postoperative neuroinflammation and memory decline could be prevented by blocking HMGB1 [177]. Similarly, elevations of plasma HMGB1 levels in rats were associated with memory deficits and anxiety after a partial hepatectomy [179]. Systemic anti-HMGB1 antibody treatment exerted neuroprotective effects and prevented the development of POCD [179]. 

Animal [176,181,182,183] and human studies [8,118,184,185,186,187,188,189] provide evidence for the role of systemic proinflammatory cytokines, such as IL-1β, TNF-α, and IL-6, in PND development. In a mouse model of tibial fracture, IL-1β [182], IL-6 [181,184], and TNF-α [176] were correlated with cognitive decline after surgery. Peripheral blockade of proinflammatory cytokines was able to mitigate cognitive deficiencies. Liu et al. [188], by conducting a meta-analysis of 54 clinical observational studies, found significantly increased peripheral C-reactive protein (CRP), IL-6, and IL-1 concentrations in POCD patients. Similarly, a recent meta-analysis of 23 randomized controlled trials showed an association of increased proinflammatory biomarkers (IL-6, IL-10, TNF-α, and CRP) with surgery and anesthesia [186].

Notably, some animal studies have found that surgery and anesthesia increased not only peripheral HMGB1 levels but also HMGB1 expression in the hippocampus in conjunction with the development of cognitive deficits [178,180]. This raises the important question of the mechanisms by which alterations in peripheral inflammatory signals can negatively impact central nervous system (CNS) function and, therefore, cognition. Peripheral proinflammatory cytokines can enter the brain and affect its function through active transport across the BBB, through leaky regions in the BBB, through activation of the vagal nerve, and through other mechanisms [190,191]. The peripheral inflammation-induced disruption of the BBB and subsequent neuroinflammation are widely accepted factors in the pathogenesis of PND.

#### 3.2.2. Disrupting the Integrity of the BBB

The BBB controls the flow of nutrients and chemical signals between the blood and the brain, thus playing a critical role in the brain’s metabolic activity and neuronal functions [192]. The BBB is composed of specialized brain microvascular endothelial cells, astrocytic foot processes, and pericytes [192,193,194]. The brain was classically thought to be an immune-privileged organ due to the existence of a BBB impervious to immune response factors [192,193,194,195,196,197]. However, the recent discovery of interactions between the CNS and the peripheral immune system has drastically changed this view [198,199]. Under pathological conditions, peripheral proinflammatory cytokines may damage the BBB integrity via the upregulation of cyclooxygenase-2 (COX-2) and matrix metalloproteins (MMPs). Activated COX-2 increases prostaglandin synthesis leading to alteration in the BBB permeability [197], while MMPs may affect the BBB permeability through the degradation of extracellular matrix proteins [198].

Considerable evidence from animal experiments supports the notion that surgery and anesthesia can compromise the integrity of the BBB [183,199,200,201,202,203,204,205,206]. Disruption of the BBB, migration of BMDMs into the hippocampus, and subsequent cognitive decline were found in a mouse model of tibial fracture [200,206]. Preoperative depletion of BMDMs prevented hippocampal neuroinflammation and memory dysfunction, suggesting the role of BMDMs in the development of POCD [183]. The role of MMP-9 and MMP-2 in surgery-induced BBB disruption and cognitive decline has been uniformly reported in several preclinical models of POCD [202,202,203]. It was further validated that *Mmp9*^−/−^ mice exhibited better cognitive performance after surgery compared to wild-type mice [204]. In addition, anesthesia and surgery were able to reduce the levels of tight junction protein in brain capillaries, increasing BBB permeability [183,199,200,202,205]. Laparotomy under isoflurane anesthesia increased mouse BBB permeability and cognitive impairment in an IL-6-dependent and age-associated manner. These events were accompanied by decreased tight junction proteins claudin, occludin, and ZO-1 [205].

Markers supporting the BBB disruption associated with POCD were also found in human patients [207,208,209,210]. Serum levels of CNS-specific proteins S-100β and neuron-specific enolase (NSE) were measured in a cohort of 15 patients before and after coronary artery bypass grafting. Serum levels of S-100β and NSE were both significantly increased postoperatively [209]. A significant correlation was found between cognitive dysfunction and an increase in the NSE level, suggesting that BBB breakdown is relevant to the occurrence of POCD. The BBB disruption was further demonstrated by using dynamic contrast-enhanced magnetic resonance imaging (MRI), which showed an increase in the permeability constant (K^trans^) after coronary artery bypass grafting. The MRI findings correlated with POCD symptoms [210]. 

#### 3.2.3. Activation of Microglia

Microglia, which account for 5 to 10% of CNS cells, are the resident brain macrophages and highly dynamic surveyors of brain parenchyma [211,212]. Microglia have emerged as important contributors to the homeostasis of the brain microenvironment. However, pathological conditions such as inflammation can result in undesirable microglial activity [211,212,213,214,215]. Activated microglia secrete proinflammatory cytokines and recruit more BMDMs into the CNS, escalating an inflammatory cascade [211,212,213,214,215].

The findings of preclinical studies demonstrate the role of microglial activation in neuroinflammation associated with anesthesia and surgery, as well as the subsequent development of PND [179,216,217,218]. The common characteristics of activated microglia, described across PND models, are upregulated expressions of classic markers such as ionized calcium-binding adaptor molecule (IBA1) and CD11b, as well as changes in cellular morphology [216,217,218]. Recent advancements in technology also provide opportunities to assess the role of microglia in vivo [219,220]. By inhibiting the colony-stimulating factor 1 receptor (CSF1R), perioperative microglial depletion alleviated neurobehavioral deficiencies in a mouse model of POCD. This protection was associated with reduced hippocampal inflammatory mediators and abrogation of hippocampal BMDM recruitment [221]. Notably, GA itself may also dramatically alter the function of microglia, causing neuroinflammation and subsequent behavioral deficiencies [222,223,224,225,226]. In this regard, two-photon microscopy imaging showed that isoflurane, but not ketamine, potentiated microglia responses to damage in mice [226]. However, clinical studies investigating the role of microglia in PND development are limited. A recent clinical observational study of eight patients undergoing prostatectomy reported a transient change in the expression of translocator protein (TSPO) in the brain’s grey matter [227]. The expression of TSPO, whose changes reflect changes in glial cell activity, decreased at 3 to 4 days postoperatively but recovered to its pre-surgery level 3 months later. Interestingly, performance during some cognitive tests correlated with TSPO levels as measured over the course of the study [227]. 

Potentially, microglia may contribute to PND development through interaction with astrocytes. Astrocytes play an important role in the maintenance of neuronal homeostasis but undergo a profound transformation termed “reactive astrocytosis” upon brain injury [228,229]. A subtype of such reactive astrocytes, called A1 astrocytes, can be induced by activated microglia through the release of the proinflammatory cytokines IL-1α, TNF, and C1q [230,231,232,233]. The A1 reactive astrocytes induce neuronal and oligodendrocyte death during injury and are present in many human neurodegenerative diseases [234,235,236,237]. In addition, A1 reactive astrocytes may further potentiate the inflammatory response by activating microglia through the release of chemokine C-C motif ligand 2 (CCL2) [also known as monocyte chemoattractant protein-1 (MCP-1)] and the activation of its receptor C-C motif chemokine receptor 2 (CCR2) in microglia [238]. This mechanism may be involved in the mediation of the neurobehavioral effects of GAs. For example, 18-month-old senescent mice exhibited behavioral deficits 1 week and 3 weeks after receiving a sedative dose of etomidate. The evidence of activated hippocampal microglia and an A1-specific astrocyte response were also found 1 week and 3 weeks after sedation with etomidate, respectively [225]. Inactivation of microglia prior to sedation reduced behavioral deficits and A1 reactive astrocyte levels [225]. The association of astrocytic activity with POCD has also been reported in human patients. Rappold et al. [239] found an inverse relation between increased postoperative concentrations of glial fibrillary acid protein (GFAP), a specific marker of astrocyte activation measured in plasma after shoulder surgery, and cognitive abnormalities identified 1 month later.

More laboratory and clinical studies are needed to clearly determine the relationship between peripheral and CNS inflammation and the development of PND. Although surgery and anesthesia-related neurocognitive abnormalities are associated with an increase in peripheral and CNS inflammatory markers, surgery/anesthesia-inflicted inflammatory responses are largely transient and revert toward baseline within 12 to 96 h after surgery [186]. A transient inflammatory response that involves the recognition and elimination of a threat and initiation of repair mechanisms is a vital part of the physiological processes that maintain homeostasis. On the other hand, excessive and especially chronic systemic inflammation lowers the threshold for many pathophysiological conditions, including neurodegenerative disorders [98,190,191,240,241]. Preexisting neurodegenerative disorders and advanced age, which tend to worsen such disorders, are among the most consistently reported risk factors for the development of PND [1,2,3,4,5,6,7,8,9,10,14,15,16,17,18,19,20,21,22,23,24,25,26,27]. It is possible that in order for the GA/surgery-induced transient stress and inflammation to be able to induce the lasting neurocognitive abnormalities of PND, they need to be primed by stress and inflammation effects of preexisting or accompanying pathophysiological conditions. Therefore, it is important to investigate pre- and post-surgery trajectories of stress and inflammation marker levels and neurocognitive function in patients.

### 3.3. Interaction of Stress and Inflammation 

The HPA axis and immune defense systems closely interact through a negative feedback loop between the systems to maintain an organism’s homeostasis [81]. Proinflammatory cytokines stimulate HPA axis activity and the production of GCs, while GCs downregulate immune system functioning, leading to the resolution of the inflammatory response and restoration of homeostasis [242]. However, elevated levels of GCs may also be accompanied by increased proinflammatory activity [243]. The possibility of simultaneous activation of both stress and immune systems with a resultant negative impact on brain physiology has been extensively studied in mood disorders [244,245]. Clinical studies found elevated levels of proinflammatory cytokines, such as IL-1β, IL-6, and TNF-α, in patients with major depressive disorder (MDD) [244,245]. Proinflammatory cytokines are believed to play a mediating role in the pathophysiology of stress-induced depressive disorders [245]. 

Transient stressful experiences occurring in everyday life, as benign as taking a test, may also increase inflammatory markers [246,247]. For example, a meta-analysis of 34 studies that measured circulating inflammatory markers and 15 studies that measured stimulated production of inflammatory markers before and after exposure to a laboratory challenge found transient increases in circulating IL-6, IL-1β, IL-10, TNF-α, and stimulated IL-1β, IL-4, and interferon-γ in response to acute stress that peaked within 2 h [248]. Notably, the effects of stress on the immune response, depending on the individual, range from a profound to little or no response [249,250]. The findings suggest that such differences in resiliency predict susceptibility to subsequent diseases [251,252]. Excessive elevations in GC levels may make the CNS immune system more permissive to inflammation later on, for example, by priming the reactivity of microglia [253,254]. Thus, stress caused by inescapable electric tail shock [253,254] or the administration of exogenous GCs [255] potentiated the activation of hippocampal microglia caused by the administration of lipopolysaccharide (LPS). Pretreatment with the GC receptor antagonist RU486 or removal of the adrenal glands prevented excessive activation of the hippocampal microglia by LPS [256]. It will be important to investigate whether individuals with greater stress/inflammation reactivity have a prior history of exposure to excessive stress and/or inflammation. Simultaneous increases in HPA axis activity and inflammation markers, especially those associated with excessive or chronic stress, may lead to accelerated cellular aging and increased vulnerability to various disease states, including age-related abnormalities such as neurodegenerative and neurocognitive changes [97,257].

Although clinical and laboratory studies provided evidence that preexisting neurodegenerative diseases and advanced age—which are typically accompanied by dysregulated stress response systems and inflammation—are contributing factors to the development of PND [1,2,3,4,5,6,7,8,9,10,14,15,16,17,18,19,20,21,22,23,24,25,26,27], we are not aware of clinical studies specifically designed to investigate the role of the interplay between preexisting stress and inflammation in the development of PND. A beautiful demonstration of the interaction of preexisting stressful experiences, inflammation, and epigenetic modifications in a rat model of POCD was made by Zhu and colleagues [258]. Specifically, they tested whether early life stress caused by maternal separations sensitized to neurobehavioral abnormalities induced by SEVO anesthesia in adulthood [258]. They found that rats neonatally subjected to maternal separation exhibited not only significantly exacerbated behavioral/cognitive impairments but also that behavioral deficiencies were accompanied by a significantly enhanced release of cytokines and the activation of intracellular NF-κB in hippocampal astrocytes [258]. Furthermore, rats neonatally subjected to maternal separation exhibited significant reductions in the expression of hippocampal GR [258]. Considering that the GR level and function are important factors in the regulation of the inflammatory response, the authors speculated that the downregulation of GRs caused by neonatal maternal separation contributes to neurocognitive deficiencies induced by SEVO exposure as adults, presumably through hypermethylation of the GR gene and a subsequent increase in proinflammatory cytokines [258]. Notably, this study found that DNA methylation and histone acetylation may act in concert to regulate GR expression. Thus, the histone deacetylase inhibitor trichostatin A, administered to adult rats, reduced GR methylation level and increased GR expression in rats subjected to early-life maternal separation and reduced inflammatory responses and behavioral deficiencies in those rats caused by exposure to SEVO [258]. These findings raise the possibility of a complex interplay between preexisting stress, inflammatory responses, and epigenetic mechanisms in the development of PND.

Considering an interplay between the stress and immune response systems, the interaction between preexisting stress/inflammation and stress/inflammation caused by GA/surgery may both be necessary for the development of PND, at least in its long-term form. This concept is, however, not supported by findings in healthy laboratory animals, who can develop significant neurobehavioral and neuroendocrine abnormalities after undergoing surgery under GA, or after exposure to GAs alone. In general, laboratory rodents seem more vulnerable to long-term neurocognitive effects of GAs than human patients. Aside from typically prolonged anesthesia exposure in relation to life span in laboratory rodents when compared to that of human patients, laboratory rodents may be more vulnerable to GAs because of persistent stress caused by confinement to cage housing. Laboratory rodents also have limited exposure to social interaction, which is typically restricted to interaction with one cagemate. In support of this contention are experimental findings that environmental enrichment alleviated neurodevelopmental abnormalities induced by early life exposure to GAs [259,260,261]. Environmental enrichment is part of life for most human patients; therefore, GA/surgery may have minimal long-term effects in human patients who experience only GA/surgery-induced transient stress/inflammation. This is one of the explanations why relatively short anesthesia exposures in children undergoing elective surgeries do not induce neurocognitive abnormalities [262,263,264,265]. On the other hand, detrimental neurocognitive outcomes are associated with repeated anesthesia exposures, which are typically required for the treatment of very sick children [266,267]. Therefore, regardless of age, human patients with pathophysiological conditions involving dysregulated stress response systems, inflammation, and neurodegenerative changes may be at increased risk of developing PND after undergoing surgery under GA, or after exposure to GAs alone. Emerging data from laboratory and clinical studies suggest that epigenetic mechanisms may play a crucial role in anesthesia, surgery-induced stress and inflammatory responses, and neurocognitive abnormalities that constitute PND [268,269].

### 3.4. Epigenetic Mechanisms

Epigenetic modifications regulate gene expression without altering DNA sequence [270]. The most extensively studied epigenetic mechanisms are DNA methylation, histone modifications, and changes in levels of noncoding RNAs (ncRNAs) [270]. Epigenetic modifications can persist for years and affect all body processes and functions, including neurogenesis, neuroplasticity, cognition, memory, and behavior [271,272,273,274,275]. Various environmental factors, such as stress, inflammation, endocrine-disrupting chemicals, alcohol, substances of abuse, anesthetic agents, and others, can initiate epigenetic modifications [271,276,277,278,279,280,281,282,283,284,285,286,287,288,289]. Furthermore, the frequency of epigenetic modifications may accelerate as people age—a phenomenon known as epigenetic drift [290,291]. Therefore, it is not surprising that an increasing number of laboratory and clinical studies find evidence for the involvement of epigenetic processes in the genesis of PND [269], for which stress, inflammation, and advanced age are among the risk factors [1,2,3,4,5,6,7,8,9,10,14,15,16,17,18,19,20,21,22,23,24,25,26,27]. Importantly, some epigenetic modifications can be transmitted from parents to the next generation via germ cells, raising the possibility of heritability of PND [34,35,36,37,39,44,48,54,58,70,292,293,294,295,296,297,298,299,300].

#### 3.4.1. DNA Methylation

DNA methylation is the biological process of converting cytosine to 5-methylcytosine (5mC) via the transfer of a methyl group (-CH3) from S-adenosyl methionine to the fifth carbon position of a cytosine ring in a cytosine-phosphate-guanine (CpG) dinucleotides site [301,302]. The -CH3 transfer is catalyzed by DNA methyltransferases (DNMTs). DNMTs form a class of enzymes that differ in their roles. DNMT3a and DNMT3b can establish a new methylation pattern by methylating unmodified DNA, i.e., they function as de novo DNMTs. DNMT1 recognizes hemimethylated DNA formed during replication and copies the original DNA methylation pattern onto the newly synthesized daughter strand (maintenance DNMT) [301,303,304,305,306]. 5mC can be oxidized into 5-hydroxymethylcytosine (5-hmC) by ten-eleven translocation (TET) enzymes. DNA methylation in different genomic regions may exert different influences on gene expression. DNA methylation, found in CpG islands (CGIs) in the promoter regions of a gene, is often associated with the suppression of the transcription of the corresponding gene [302,306,307]. The balance between DNA methylation and demethylation is important for normal brain development and function [308,309,310].

Investigation of the role of DNA methylation in PND pathogenesis in human patients is in its initial stage. A recent clinical study evaluated memory, learning, attention, executive functions, cognitive flexibility, and global DNA methylation in 124 patients older than 65 years before and 7 days after hip replacement surgery. 5-mC level in peripheral leukocyte DNA was used as a marker of global DNA methylation. One week postoperatively, only patients who exhibited neurocognitive deficiencies (19.4%) had reduced peripheral leukocyte 5-mC levels. Preoperative levels of 5-mC were similar in patients who were and were not affected cognitively [311]. Sadahiro and colleagues also used peripheral blood, more specifically mononuclear cells, collected from elderly patients to assess genome-wide DNA methylation patterns prior to major surgery, immediately after surgery, and at discharge from the hospital [312]. They found differentially methylated regions in genes associated with the immune system, and many of these altered patterns persisted until hospital discharge. These DNA methylation patterns were similar, but not identical, in patients who underwent different types of major surgeries, allowing us to speculate that not just a particular pathophysiological condition or surgical disease but rather surgical injury, anesthesia, stress, and inflammation, in general, could be important causes of the observed epigenomic modifications.

Animal studies with SEVO and isoflurane support the possibility that GAs themselves may induce DNA methylation changes in the brain and neurobehavioral deficiencies. In one such study, 18-month-old mice were exposed to 2% SEVO for 2 h [313]. The animals’ behavior was evaluated until day 7 after exposure to SEVO, followed by the collection of brain tissue. The SEVO-induced behavioral deficiencies were accompanied by a decrease in global DNA 5hmC levels, with the hippocampus and amygdaloid nuclei showing the greatest declines [313]. Furthermore, profound neurobehavioral and epigenomic changes were found in rats neonatally exposed to GAs. For example, neonatal SEVO exposure increased the expression of hippocampal and hypothalamic DNMTs and the methylation of the brain-derived neurotrophic factor (*Bdnf*) gene and downregulated expression of *Bdnf* and neuron-specific *Kcc2* Cl^−^ exporter genes [59,132,314]. In adulthood, these animals exhibited exacerbated corticosterone responses to stress, changes in hippocampal synaptic morphology, and behavioral abnormalities. The DNMT inhibitor decitabine, administered prior to exposure to SEVO, prevented the above-mentioned effects of the anesthetic [59,132,314]. In addition, Wu and colleagues [315] reported that neonatal isoflurane exposure increased the expression of hippocampal DNMT1 and upregulated the methylation level of *Bdnf* exon IV, which was accompanied by decreased mRNA and protein levels of *Bdnf*. Another recent study revealed that repeated exposure of neonatal rats to SEVO resulted in impaired behavior and increased and reduced transcriptions of hippocampal *Dnmt1/3a* and TET1, respectively [316]. These rats had corresponding higher and lower levels of 5-mC and 5-hmc, respectively, as well as hypermethylated synaptic protein genes whose expression was reduced. The authors concluded that epigenetic mechanisms were involved in the mediation of the behavioral effects of SEVO, as such effects were alleviated by pretreatment with the inhibitor of DNMTs, 5-aza-2-deoxycytidine [316].

The analysis of genome-wide DNA methylation patterns in the hippocampus of adult rats neonatally exposed to SEVO identified 407 differentially methylated regions in 391 genes (differentially methylated genes) [317]. The overrepresentation analysis of the differentially methylated genes yielded multiple enriched gene ontology (GO) terms that covered a wide array of biological processes ranging from nervous system development and function to regulation of stress responses, metabolic processes, vasculature development, and blood circulation [317]. By acting via mechanisms that involve epigenetic changes and dysregulation of stress response systems, these findings raise the possibility that early life exposure to GAs may predispose to PND, induced by subsequent exposure to surgery and/or GAs in adulthood. In fact, we received experimental evidence that early-life anesthesia exposure may sensitize rats to the adverse effects of subsequent exposure to SEVO [132]. Thus, rats neonatally exposed to SEVO responded to subsequent SEVO exposure on P19–P21 with EEG-detectable seizures, stress-like corticosterone secretion, and altered expressions of both *Kcc2* and *Dnmt* genes in the cortex as well as the hypothalamus. The effects of neonatal exposure to SEVO were mitigated by pretreatment of neonatal rats with the DNMT inhibitor decitabine administered prior to exposure to the anesthetic [132].

#### 3.4.2. Histone Modifications

Posttranslational modifications (PTMs) of histones are another critically important regulator of chromatin structure and gene expression [318]. Within a cell nucleus, an octamer of histone proteins (2 of each H2A, H2B, H3, and H4) is encircled by 147 base pairs of DNA to form a nucleosome, the fundamental unit of chromatin. The N-terminal tails of histone proteins contain multiple sites for enzyme-catalyzed modifications, such as acetylation, methylation, phosphorylation, ubiquitination, glycosylation, and ADP-ribosylation [318,319,320].

Among these modifications, histone acetylation is the most widely studied histone modification in the field of PND. Histone acetyltransferases (HAT) catalyze the transfer of acetyl groups from acetyl coenzyme A to lysine (K) residues of histone tails. Histone deacetylation is catalyzed by histone deacetylases (HDACs) [318,319,320]. In contrast to DNA methylation, which reduces gene transcription, histone acetylation promotes gene transcription by relaxing the charged attraction between histones and DNA, thereby permitting access of transcription factors or RNA polymerase to DNA [318,319,320]. A number of studies reported that the dysregulation of hippocampal histone acetylation might contribute to surgery-induced cognitive impairments, whereas restoration of histone acetylation by HDAC inhibitors or environmental enrichment may exert neuroprotective effects [268,321,322,323]. For example, Jia and colleagues found that an exploratory laparotomy in 16-, but not in 3-month-old, mice impaired hippocampus-dependent long-term memory. The memory impairment was accompanied by decreased acetylation of hippocampal histone H3 and H4 and decreased expression of synaptic plasticity-related proteins [323]. Treatment with the histone deacetylase inhibitor suberoylanilide hydroxamic acid (SAHA) alleviated both behavioral and molecular effects of surgery, including restoration of histone acetylation levels and increase in the expression of hippocampal BDNF, synapsin 1, and postsynaptic density 95 [323]. Not only surgery under GA but also GAs themselves may induce alterations in histone acetylation and neurocognitive abnormalities. For example, 22-month-old mice exposed to isoflurane exhibited impaired spatial learning and memory, reduced acetylation of hippocampal histone H3 lysine 9 (Ac-H3K9) and histone H4 lysine 12 (Ac-H4K12), increased the levels of histone deacetylases 2, downregulated expression of hippocampal BDNF, downregulated BDNF-TrkB signaling pathway, and elevated levels of proinflammatory cytokines IL-2, IL-4, and IL-10 [22,324]. Similarly, repeated exposure of aged mice to SEVO resulted in cognitive deficits and reduced histone acetylation in the hippocampus [325].

Histone methylation (i.e., the addition of methyl groups to the N-terminal tail of histone proteins catalyzed by histone methyltransferases (HMTs)) has been linked to either transcriptional activation or repression, depending on the specific methylated residues (lysine or arginine) and the number of methyl groups added (mono, di, or trimethylation) [318,319,320,326]. Lysine trimethylation at the ninth position of histone H3 (H3K9me3) is one of the most studied histone methylations in the process of memory formation and correlates with gene silencing [327,328]. A recent study explored its association with PND. A laparotomy under isoflurane anesthesia in adult male mice increased the binding of H3K9me3 to the *Bdnf* exon IV promoter and decreased *Bdnf* expression in the dorsal hippocampus. The histone methyltransferase antagonist chaetocin restored BDNF expression and reversed the fear memory deficits induced by anesthesia and surgery [329].

#### 3.4.3. Noncoding RNA

Noncoding RNAs are a vast and diverse family of non-protein-coding transcripts that regulate gene expression at the levels of transcription, RNA processing, and translation [330,331]. The most widely studied and best-characterized ncRNAs are microRNAs (miRNAs), long noncoding RNAs (lncRNAs), and circular RNAs (circRNAs) [330,331,332,333]. Mature miRNAs span 19 to 25 nucleotides and act to silence gene expression by binding to complementary 3′ untranslated regions (UTR) of target messenger RNA (mRNA) to inhibit mRNAs translation and accelerate their degradation. One miRNA can bind to many target mRNAs, and conversely, each target mRNA can be regulated by multiple miRNAs [334,335]. Compared with other organs, the brain has a particularly high percentage of miRNAs [336,337]. The crucial roles of miRNAs in cognitive functions and in the development of neuropsychiatric disorders have been demonstrated by multiple studies [338].

LncRNAs are defined as transcripts exceeding 200 nucleotides and have been shown to control diverse cellular processes, such as transcriptional regulation, organization of nuclear domains, and cell cycle regulation [339]. Changes in the expression of lncRNAs are associated with cancer and several neurological disorders [340,341]. Circular RNAs are covalently closed single-stranded noncoding RNAs. Many circRNAs exert important biological functions by acting as microRNA or protein inhibitors (“sponges”) [342]. Furthermore, circRNAs have been implicated in diseases such as neurological disorders, cardiovascular diseases, and cancer [343,344,345]. More recently, evidence of the involvement of ncRNAs in the pathogenesis of PND was reported [345,346]. Gao et al. performed a circRNA microarray to screen for differentially expressed circRNAs in POCD patients and investigated their potential role in the development of POCD. The data showed 210 differentially expressed circRNAs, 133 upregulated and 77 downregulated in POCD patients when compared with non-POCD patients. Differentially expressed circRNAs were associated with the regulation of cell adhesion and nervous system development [347]. Most studies investigate the changes of a specific ncRNA and its target gene expression in POCD animal models. See Yazit et al. [348] and Yang et al. [349] for recent comprehensive reviews of ncRNAs in PND.

### 3.5. Intergenerational PND

An increasing number of human studies report evidence for heritable effects of parental stress/trauma, inflammation, drug misuse, endocrine-disrupting chemicals, obesity, and others despite the confounding effects of social, cultural, educational, and behavioral factors that complicate the investigation of parental experience-induced changes in an offspring’s development and resilience [37,38,39,40,41,42,43,44,45,46,47,48,49,50,51,52,53,54,55,61,62]. Heritable adverse effects of parental stress have been extensively studied, partially because in specific circumstances, such as war or famine, large groups of people in relatively compact living areas within a defined period of time were affected [32,36,63,64,65,66,67,68,69,70]. Although the exact mechanisms mediating the effects of parental experiences on offspring in humans remain to be elucidated, animal and human studies support the involvement of epigenetic reprogramming [34,35,36,37,39,44,48,54,58,70,292,293,294,295,296,297,298,299,300]. Such epigenetic reprogramming includes changes in DNA methylation, histone modifications, and noncoding RNAs [34,35,36,37,39,44,48,54,58,70,292,293,294,295,296,297,298,299,300]. The relatively long-lasting stability of DNA methylation within 5′ CpG sequences support their role in the mediation of persistent and heritable effects of environmental stressors [55]. This role is further supported by the fact that methylation of the *GR* (*NR3C1*) gene is one of the most extensively studied processes in stress- and inflammation-induced heritable neuropsychiatric disorders in humans [36,53,55,66,69,70,350,351,352]. Changes in *NR3C1* methylation, particularly in peripheral blood monocytes (PBMCs) that correlate with the severity of neuropsychiatric symptoms are found in parents and offspring with post-traumatic stress disorder (PTSD), borderline personality disorder, MDD, and other disorders [36,53,55,66,69,70,350,351,352]. Interestingly, findings in human studies demonstrate that distinct changes in *NR3C1* methylation patterns in offspring may occur depending on whether paternal or maternal stressful experiences lead to parental PTSD [36,69]: specifically, maternal, but not paternal, PTSD-induced HPA axis biomarkers of PTSD in offspring. However, where both parents had PTSD, the offspring were more likely to have symptoms of dissociative amnesia [36]. Notably, human studies of the heritable effects of stressful parental experiences suggest that abnormalities in offspring can be detected even if their mothers were not overtly symptomatic [353]. An involvement of DNA methylation in heritable effects of parental stress is also supported by findings of Yehuda and colleges that Holocaust survivors and their offspring have altered methylation of the stress regulatory *FKBP5* gene [354]. The *FKBP5* gene methylation and resulting changes in the synthesis of the FK506 binding protein 51 play an important role in the GR-mediated regulation of the HPA axis response to stress [355,356].

Despite accumulating evidence that stressful parental experiences prior to conception may be passed to future offspring, how these experiences transmit and result in psychologically vulnerable phenotypes in offspring remains largely unknown. Studies in animal models have shown that parental treatments with GCs prior to conception result in epigenetic changes in spermatozoa and phenotypic alterations in offspring similar to those induced by corresponding parental stress [296,357,358,359,360]. In addition, treatment of parents with GR antagonists prior to stress exposure ameliorated the heritable effects of those stressful experiences [358], suggesting that GC-induced changes in parental germ cells are involved in the heritable effects of stressful parental experiences. The mechanisms that link parental exposure to GCs and changes in gamete epigenetic alterations require further investigation. Given that expression of the GR has been reported in primary spermatocytes and oocytes [361,362,363], it is plausible that GCs directly activate gamete GR to modulate the epigenetic landscape, affecting the offspring’s phenotypes [358,359].

The GC corticosterone and the GC receptor gene (*Nr3c1*) may also be involved in the initiation and mediation of intergenerational PND in rats [57,58,59]. First, using a rat model, we have demonstrated that neonatally SEVO-exposed rats (F0) and their unexposed male, but not female, progeny (F1) developed similar, but not identical, neurobehavioral abnormalities [58]. The exposed F0 adults exhibited behavioral deficiencies, exacerbated HPA axis (corticosterone) responses to stress, and impaired expression of hypothalamic *Kcc2,* with males more severely affected than females. Interestingly, neurobehavioral phenotypes in F1 offspring differed depending on whether one or both parents were neonatally exposed to SEVO. The F1 males of both exposed parents were the only group of F1 males that had a significantly impaired expression of hippocampal *Kcc2.* They were the only group that exhibited significant abnormalities in spatial memory during the Morris water maze test. On the other hand, the male progeny of exposed sires/control mothers were the only group that exhibited significant alterations in the elevated plus maze (EPM) and PPI of acoustic startle responses [58]. In contrast to exposed F0, their progeny exhibited normal corticosterone responses to stress. Bisulfite sequencing revealed increased CpG site methylation in the *Kcc2* promoter in F0 sperm and F1 male hippocampus as well as the hypothalamus, concordant with changes in *Kcc2* expression in specific F1 groups, supporting the involvement of epigenetic modification of *Kcc2* in transmitting/mediating the effects of neonatal SEVO exposure in the next generation [58]. In support of the involvement of corticosterone in intergenerational effects of neonatal SEVO exposure, the NKCC1 inhibitor bumetanide alleviated the SEVO-increased corticosterone secretion in exposed P6 rats [126] and prevented abnormally expressed genes in the hypothalamus of their male offspring (Figure 2, authors’ unpublished observations). Notably, neither paternal neonatal exposure to SEVO nor pretreatment with bumetanide prior to SEVO exposure affected the expression of these genes in female offspring (Figure 2, authors’ unpublished observations).

Remarkably, stress-like intergenerational effects of SEVO may not be limited to parents exposed at neonatal age. Thus, in a study in which male and female Sprague-Dawley rats (F0) were exposed to SEVO on 3 alternating days in young adulthood (P56) and mated 25 days later to generate offspring (F1), both generations were affected [57]. Specifically, F0 male and female rats exposed to SEVO had high corticosterone levels at the time of exposure, but only F0 males exhibited persistent neurobehavioral deficiencies [57]. Still, F0 sires and dams had similar alterations in DNA methylation patterns of the *Kcc2* gene in sperm and ovarian tissue, respectively. Even though only F0 males were significantly affected in the long-term, both F0 males and females passed neurobehavioral abnormalities to F1 male offspring but not females [57]. These findings suggest that SEVO-induced acute corticosterone secretion at the time of exposure is involved in initiating epigenetic reprogramming of F0 male and female gametes and subsequent intergenerational abnormalities.

The F1 male offspring of parents exposed to SEVO neonatally or in young adulthood exhibited similar neurobehavioral deficiencies [57,58]. These findings are in line with similar heritable effects of stress exposure at different times of parental life but may be inconsistent with the findings of some human studies [364,365]. Thus, epidemiological data from Swedish famine periods suggest that there may be specific developmental windows during which germ cells are more vulnerable to this type of environmental influence. Boys exposed to famine before puberty were more likely to have children and grandchildren with an increased risk of cardiovascular disease and diabetes than males exposed outside this period [364,365]. This disparity in the timing of germ cell vulnerability between our and other laboratories’ findings in rodents and the findings from the Swedish cohorts may be dependent on species, timing, or type of stressors. Our findings in rats also point to a wide range of intervals (neonatal exposure vs. exposure in young adulthood) between SEVO exposure and subsequent mating, within which SEVO can induce intergenerational effects. Of potential relevance, human studies report an increased prevalence of neuropsychiatric disorders in children conceived decades following the Holocaust [36]. Therefore, more specifically designed studies are needed to determine heritable outcomes depending on the age at the time of stressful experiences, type of stress, and delays between stress and conception. The heritable effects of SEVO in healthy laboratory rodents were also confirmed by other groups [56,60].

General anesthesia is rarely required for healthy humans. Therefore, an important question is whether the stress and inflammation effects of accompanying surgery or pathophysiological conditions interact with GAs to result in more profound intergenerational PND. An example of such a pathophysiological condition could be a traumatic brain injury [366,367,368,369,370]. Traumatic brain injury, with >50 million cases per year, is a dominant cause of disability in young adults [371,372,373], many of whom will have children later. The pathophysiology of TBI involves excessive dysregulation of stress response systems, neuroinflammation, and cognitive decline [366,368,369,372,373,374,375,376,377,378,379,380,381]. Patients with a history of TBI frequently require general anesthesia/surgery or sedation to treat conditions unrelated to TBI or injuries that may occur at the time of TBI. More profound intergenerational PND in individuals with TBI is supported by a recent study in rats [128]. In this study, young adult male rats (F0) were subjected to surgery (as part of the procedure to inflict TBI), a midline fluid percussion-induced moderate TBI [369], and subsequent exposure to SEVO to model sedation that could be required for management of conditions unrelated or related to TBI (TBI plus SEVO rats). Three weeks later, F0 male rats were mated with control females to generate offspring. The F0 TBI plus SEVO rats, when compared to F0 rats exposed to SEVO only or SEVO and surgery, exhibited greater acute and persistent dysregulation of the HPA axis, increased inflammatory markers, and behavioral deficits (an animal model of PND) [128]. All these abnormalities were also more profound in their subsequent offspring (F1). Similar to the intergenerational effects of SEVO by itself, only F1 males were affected (intergenerational PND) [128]. Overlapping hypermethylated *Nr3c1* CpG sites in the spermatozoa of sires and in the hippocampus of male, but not female, offspring (especially in the proximal promoter) reduced *Gr* expression in the F1 male (not female) hippocampus and exacerbated GR-dependent HPA axis responses to stress in F1 males (not females) support the involvement of stress-initiated epigenetic alterations in the intergenerational transmission of paternal surgery/TBI/SEVO-induced PND. In addition, SEVO and surgery together induced, in general, more prominent intergenerational changes in stress responses, inflammation, and behavior when compared to those induced by exposure to SEVO only. The findings of this study in rats not only provide evidence of an interaction of paternal surgery, TBI, and exposure to SEVO to induce intergenerational PND but also add to the potential complexity of identifying intergenerational PND in humans by demonstrating that in some conditions, offspring could be affected even when their exposed sires did not exhibit detectable deficiencies [57,353]. The findings that *Nr3c1* was similarly hypermethylated and exhibited similarly reduced expression in the brains of F0 TBI plus SEVO sires and that both generations exhibited similar neuroendocrine, inflammatory, and neurobehavioral abnormalities point to similar initiating mechanisms of the F0 somatic and germ cell effects of this combined insult (surgery, TBI, and SEVO), with dysregulated stress response systems and elevated levels of corticosterone as plausible mediators.

An important difference between the intergenerational effects of SEVO alone [57,58,59] and the combination of surgery, TBI, and SEVO [131] was an increase in inflammation markers in the TBI plus SEVO sires and their offspring. The interaction between acute stress- and inflammation-like effects of SEVO, surgery, and TBI may lead to the observed greater abnormalities in both generations. The interaction of the effects of paternal surgery, TBI, and SEVO to induce dysregulation of HPA axis functioning in F0 sires and F1 male offspring was especially evident in the reduced expression of both hypothalamic and hippocampal corticoid receptor genes and increased corticosterone responses to stress. Dysregulated stress response systems and inflammation play key roles in the pathogenesis of neurodegenerative/neurocognitive disorders and aging [106,204,240,382,383,384,385]. Hence, these findings further support the possibility that in young adult males, the adverse effects of surgery, TBI, and SEVO interact to induce PND, which may further worsen as life progresses.

Because of the priming effects of excessive stress and inflammation, it will be important to test whether surgery and/or sedation do not need to be concurrent with TBI or other pathophysiological conditions to lead to cumulative or synergistic intergenerational PND. In other words, patients who experience TBI may develop PND because of later distant exposure to surgery and/or sedation. F1 males only were affected by parental surgery/TBI/SEVO or SEVO alone, suggesting similar initiating mechanisms of their intergenerational effects. Our unpublished observations indicate that F1 male, but not female, offspring of sires exposed to SEVO have altered expression of *aromatase*, estrogen receptor α (*Erα)*, and *Dnmt* 3a/b genes in the hypothalamus at birth; these effects were attenuated by pretreatment of F0 rats with bumetanide, which prevented SEVO-caused increase in corticosterone (Figure 2, authors’ unpublished observations). All these genes encode proteins that are key for male-specific biological processes during the perinatal period in rats, specifically brain masculinization initiated by organizational effects of testis-derived testosterone [386,387,388,389,390,391,392,393,394,395,396]. Considering that parental surgery/TBI/SEVO resulted in more profound F1 abnormalities, it is possible that greater parental immune/inflammatory processes may contribute to F1 sex-specific abnormalities. For example, studies in animals suggest that testis-produced testosterone, acting via the E2/ERα pathway, stimulates the production of prostaglandin and activation of microglia, which in turn regulates sex-specific neurodevelopment and behavior [386,387,388,389,390,391,392,393,394,395,396]. Future studies will be needed to test this and other mediating mechanisms.

Future work will also be needed to evaluate the effects of surgery, TBI, and SEVO in F0 females. Women represent almost half of all TBI-related emergency room visits [397,398,399]. F0 females exposed to SEVO in young adulthood are less affected neurobehaviorally than F0 males, but F1 male offspring of both F0 males and females exposed to SEVO exhibit neurobehavioral deficiencies [128], suggesting that F0 females may act as asymptomatic “silent carriers” of PND to male offspring.

The findings of heritable effects of parental stress in humans despite multiple confounding factors, along with the findings of stress- and inflammation-like intergenerational effects of surgery/TBI/SEVO in young adult male rats in this study, with its strictly controlled experimental conditions, draw attention to the need for understanding and management of PND in millions of young adult patients with TBI, and potentially in their offspring. Considering that persistent inflammation and excessive stress are associated with accelerated cognitive decline in later life [384,385,400], these findings suggest that surgery/TBI/SEVO-induced PND in the F0 sires and their F1 male offspring might worsen as life progresses. These findings also highlight the need to investigate the dynamics of intergenerational PND in human patients with TBI across the life span of several generations. Finally, the current findings suggest that blood levels of cortisol and inflammatory markers as well as some behavioral paradigms, such as prepulse inhibition of the acoustic startle response, might be used as biomarkers to objectively and safely measure heritable PND in humans.

## Figures and Tables

**Figure 1 biology-12-00567-f001:**
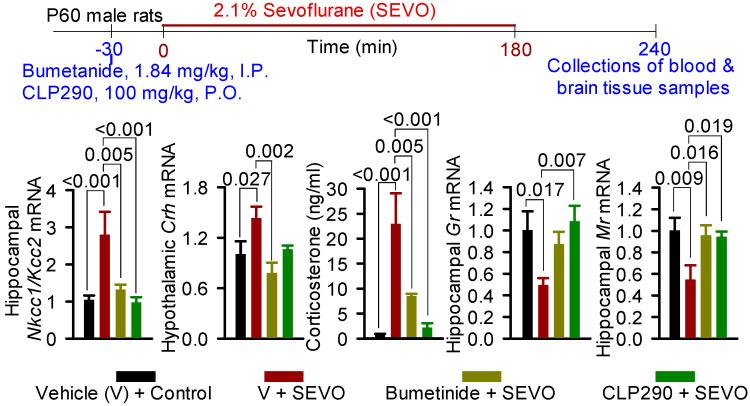
Acute stress-like effects of sevoflurane in postnatal day (P) 60 male rats. Alleviating effects of pretreatments with bumetanide or CLP290. Top panel: study design. Data are mean ± SEM of 5 rats/group. The *p*-values of multiple pairwise comparisons are shown in the respective plots above the horizontal lines. I.P.: Intraperitoneal; P.O.: Per os; *Nkcc1*: Na^+^−K^+^−2Cl^−^ gene; *Kcc2*: K^+^−2Cl^−^ gene; *Crh:* Corticotropin-releasing hormone gene; *Gr*: Glucocorticoid receptor gene; *Mr:* Mineralocorticoid receptor gene. See Appendix A for details.

**Figure 2 biology-12-00567-f002:**
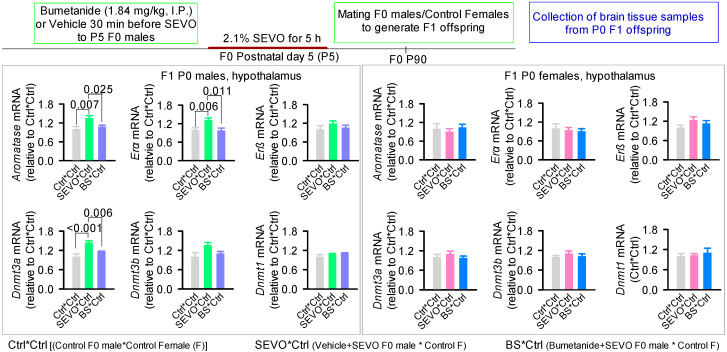
The postnatal day (P) 0 F1 males, but not females, of SEVO-exposed F0 males and Control females had abnormalities in expressions of genes that are essential for brain masculinization during a critical period of brain sexual differentiation. Bumetanide deterred F0 SEVO effects in F1 male offspring. Top panel: Illustration of experimental design. Data are mean ± SEM of 5 rats/group/sex. The *p*-values of multiple pairwise comparisons are shown in the respective plots above the horizontal lines. I.P.: Intraperitoneal; *Erα:* estrogen receptor α gene; *Erβ:* estrogen receptors β gene; *Dnmt3a:* DNA methyltransferase 3a gene; *Dnmt3b:* DNA methyltransferase 3b gene; *Dnmt1:* DNA methyltransferase 1 gene. See Appendix A for details.

## Data Availability

All original experimental data and results of respective analyses will be available from the corresponding author at reasonable request.

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
