# Peer review of "Intergenerational Perioperative Neurocognitive Disorder"

_biology, 2023, doi:10.3390/biology12040567_

Round 1
Reviewer 1 Report
Dear Authors,
Thank you for allowing me to read this interesting work.
While the title and topic of intergenerational PND are very interesting, the article can be structured better to highlight the real topic 'intergenerational PND' more.
On a side-note I fail to see what the contribution of your own research is to this review article.
Author Response
Dear Reviewer,
We appreciate your constructive review of our manuscript. We trust we have addressed all of your comments and the manuscript has been substantially improved. The changes in the text of the revised version of the manuscript are indicated using the font of red color.

Reviewer 2 Report
The paper is far too extensive to be considered as a mini-review.
I hope my comments would help improving the paper.
1.
Simple summary do not contain sufficient content of the manuscript.
There are too many references on the manuscript. Please remove unnecessary or redundant references.
2.
I fully agree that high dose or prolonged use of anesthetic drugs may induce neurotoxicity.
However the authors may overlook the organprotective effects of the anesthetic drugs.
As the authors decribed in page 5, the exogenous GCs also showed biphasic effect on POCD/delirium
3.
Part 2.1
I suggest that the contents for GABA and HPA axis are separately described. GABA is main neurotransmitter for general anesthesia. Even GABA takes influences from HPA axis, GABA independently acts in the several neuroprotective pathways.
Please add a subtopic for GABA prior the topic for HPA axis
4.
For the Figures, although it is the author's own unpublished data, the method of experiment shoud be described
at least as supplementary materials.
5.
The contents through line 647 to 670 limitedly describes GA-volatile agent. They may not for other GA-intravenous anesthetic agents. (no evidence?)
Minor concerns
line 9: ga > GA
line 10: accelerated > Accelerated
(remove bold style, Capitalize the first letter)
line 84: [40–58,64,65] ; remove bold style, correct word size
line 94: in animal studies [60–62], ; remove bold style, correct word size
line 98: revise the size of indentation
line 104: GA sevoflurane > GA-sevoflurane
line 275,278,281, 656, 684, 819, 825, 826, 827 Nkcc1/Kcc2 > NKCC1/KCC2, Kcc2 > KCC2 ; remove italic style
line 278 consider revision to "increases in both NKCC1/KCC2 ration and corticosterone release"
line 282: Crh >CRH, Gr>GR, Mr>MR; also remove italic style
line 336 Note: etomidate may affect adrenal insufficency even in the clinical settings.
line 342 ET > etomidate?
line 422 consider remove respectively unless COX-2 and MMPs do not share similar properties
line 439 -/- ; consider superscript
line 446 ~ MRI showed BBB impairment ~ ?
line 483 add description for CCR2
line 486 please remove 'the GA'
line 507 can be a part of ?
line 656, 662, 663, 730 Bdnf > BDNF; also remove italic style
line 665, 684, 935 Dnmt > DNMT; also remove italic style
line 707 but not in 3-month-old?
line 713 but also GAs themselves?
line 746 consider 'lncRNAs are ~'
line 849~903 word size corrections are required
line 914 alone [60-62]
line 915 SEVO [131]
line 955 in; remove bold style
Author Response

(The authors gave the same response as above.)

Round 2
Reviewer 2 Report
Thank you for your efforts to revising the paper.
My opinions are considered on the revised manuscript.
I understand authors' reply and confirm the manuscript is well organized.
Minor concern.
I recommend to add the descripition for supplementary material in the manuscript.
Author Response
Dear Reviewers,
We appreciate the thoughtful repeat review of our manuscript and the second chance to respond to the reviewer’s recommendation. We have addressed the reviewer’s recommendation by adding to figure legends the following statement “See Supplementary material for details”.
The changes in the text of the revised version of the manuscript are indicated using visible track changes.